# The Impact of Cognitive Reappraisal Intervention on Depressive Tendencies in Chinese College Students: The Mediating Role of Regulatory Emotional Self-Efficacy

**DOI:** 10.3390/bs15050562

**Published:** 2025-04-22

**Authors:** Ting Lu, Kejing Liu, Xiang Feng, Xinyu Zhang, Zhuang She

**Affiliations:** 1Department of Psychology, Faculty of Education, Hubei University, Wuhan 430061, China; lkj12282024@163.com (K.L.); alenmksaem@163.com (X.F.); 15983663979@163.com (X.Z.); 2Department of Psychology, School of Social and Behavioral Sciences, Nanjing University, Nanjing 210093, China; zhuang.she@nju.edu.cn

**Keywords:** college students, depressive tendencies, intervention, cognitive reappraisal, reading and writing task

## Abstract

This research was conducted to explore how a cognitive reappraisal intervention influences depression in college students with depressive tendencies, as well as to examine how regulatory emotional self-efficacy mediates this link. An unstructured questionnaire was utilized to evaluate the effects of a cognitive reappraisal intervention, while online reading and writing tasks were implemented to address the depressive symptoms among college students exhibiting tendencies. The participants in the study included 98 college students exhibiting depressive tendencies. The students’ depression, regulatory emotional self-efficacy, and cognitive reappraisal scores were measured before and after the intervention. The findings indicated that interventions focused on cognitive reappraisal can effectively reduce depression in college students exhibiting depressive tendencies. Additionally, regulatory emotional self-efficacy played a significant mediating role in the link between cognitive reappraisal intervention and depression levels among these students. The results provide valuable insights regarding interventions for depressive tendencies, highlighting their crucial role in enhancing the mental well-being of college students.

## 1. Introduction

As a common emotional disorder (mood disorder) with the main clinical manifestations of a depressive mood, the loss of pleasure, and cognitive impairment ([23]), depression poses a great threat to people’s mental health. As college students are changing from teenagers to adults, they are more likely to be impacted and more susceptible to depression ([29]). According to previous studies, depression is more common among college students than in the general population ([5]; [9]). For example, depression affects up to 24.71% of Chinese college students, and its prevalence is rising annually ([41]). Therefore, effective prevention and intervention measures to address this growing phenomenon among college students are essential.

The harmful effects of depression have long been studied, and thus many prevention and intervention studies on depression have been conducted. According to these previous studies, many types of depression intervention therapies, such as drug treatment, attention bias correction, and cognitive behavioral therapy (CBT), have been identified. Drug treatment is an important means of treating depression. However, because the use of drugs is often limited by various factors, only 40~70% of patients choose drugs as a means of treatment for their depression ([35]). The attention bias correction method involves addressing the individual attention bias through systematic standardized computer programs ([22]). However, some researchers claim that this training method lacks principles and that its scientific basis must first be investigated ([10]). CBT, which aims to intervene in the thoughts and behaviors of individuals to change their misconceptions ([13]), is a widely used traditional intervention method for depression. Many studies have demonstrated good therapeutic effects of CBT on depression intervention in college students ([4]; [49]); its use in China, however, is limited due to the substantial time commitment and its financial costs ([34]). Therefore, the development of a short, convenient intervention method or strategy that is akin to daily experience and that could enable college students to alter their feelings and thinking when they are faced with low resources and a low time investment could provide new ideas for depression intervention ([39]).

One of the core features of depression is emotional dysregulation, so it is very important to focus on a parameter that is highly correlated with improving depression during the development of intervention strategies: emotional regulation ([26]). Emotional regulation is a strategy that people use to control the production and performance of their emotions, and it consists of two main methods of emotional regulation: cognitive reappraisal (CR) and expression suppression (ES) ([18]). Cognitive reappraisal (CR) involves altering individuals’ understanding of emotion-induced situations from a cognitive perspective and thereby changing their emotional experience. Research has shown that CR is effective in reducing negative emotions in depressively inclined college students ([28]). However, expression suppression (ES) requires individuals to inhibit the expression of emotionally expressive behaviors as they occur ([18]), which is significantly associated with depressive symptoms in college students ([31]). Although ES can be able to reduce negative emotions in college students, individuals still experience a lot of negative emotions compared to CR ([14]; [44]; [45]). Moreover, research indicates that depressed undergraduates use fewer adaptive strategies, such as CR, and more negative strategies, such as ES ([21]), which makes it more difficult for individuals to recover and withdraw from negative emotions ([17]). Furthermore, undergraduates who use ES more frequently experience increased pressure to socialize with the people around them, which ultimately damages their social relationships ([30]). If interventions for depressed college students used more CR strategies, they might be able to reduce depressive tendencies, but currently there is a gap in the development of intervention strategies based on this perspective. Therefore, according to the findings mentioned, this research centers on employing emotional regulation techniques to intervene the depressive tendencies observed among college students and proposes *Hypothesis 1: CR intervention increases the individual’s tendency to use CR strategies, thus significantly reducing depression among college students*.

Although some studies have examined the effects of CR on depression in college students, the mechanisms underlying the relationship between the two remain less explored, and the underlying mechanisms need to be explored when developing depression intervention strategies ([16]; [32]). Regulatory emotional self-efficacy (RESE) refers to the individuals’ confidence in their capacity to regulate their emotional control ([6]). Research has confirmed that college students’ cognitive dispositions can positively influence RESE ([15]), and a high sense of RESE can be effective in suppressing depression ([36]). A study on mindful cognitive intervention for the negative emotions of college students showed that RESE plays a mediating role between mindful cognition and depression ([48]). We can assume that the use of cognitive reappraisal strategies can not only influence college students’ depressive tendencies but that this effect may be realized through the intermediate variable of regulatory emotional self-efficacy (RESE). Hence, this study proposes *Hypothesis 2: regulatory emotional self-efficacy serves as a mediator in the connection between CR intervention and depressive symptoms among college students.*

## 2. Methods

### 2.1. Participants

College students were recruited to complete the Beck Depression Inventory II (BDI-II) through an online platform (https://www.wjx.cn/ (accessed on 10 June 2021)). The participants could withdraw and close the questionnaire at any stage without any reason. All the participants completed the questionnaire voluntarily during the process. Finally, we gathered a total of 1002 questionnaires. After eliminating 84 invalid questionnaires with unacceptable information and/or incomplete items, we obtained 918 valid questionnaires, with an effective recovery rate of 91.6%.

We used G*Power software (https://www.psychologie.hhu.de/arbeitsgruppen/allgemeine-psychologie-und-arbeitspsychologie/gpower (accessed on 10 June 2021)) to calculate the required sample size of 54 (effect size *f* = 0.25, *α* = 0.05, 1 − *β* = 0.95). According to the calculated sample size and the BDI-II scoring standard, a score of ≥ 19 was required for participation ([25]); 98 college students were recruited from the 918 people who submitted valid questionnaires to participate in the experiment.

The participants were randomly assigned by central stratification by staff who did not participate in other parts of the experiment, and the randomization was stratified by each center with minimization for participants’ age, sex, and grade. Next, 49 (23 males; 26 females; *M* ± *SD* = 20.98 ± 1.39) participants were randomized to the intervention, and 49 (23 males; 26 females; *M* ± *SD* = 20.47 ± 1.39) participants were randomized to the control.

### 2.2. Measures

#### 2.2.1. Emotional Regulation Questionnaire

The emotional regulation questionnaire (ERQ) was used to evaluate the participants’ utilization of emotional regulation strategies ([19]). Each of the 10 items was scored on a Likert 7 scale, of which 6 items measure cognitive reappraisal (CR) and 4 items measure expressive suppression (ER). This questionnaire has good applicability in the Chinese population ([40]). In this study, the internal consistency coefficient was 0.69 for ERQ, 0.86 for CR, and 0.82 for ES. The results of confirmatory factor analysis showed that *χ*^2^/*df* = 2.14, RMSEA = 0.04, GFI = 0.95, AGFI = 0.92, TLI = 0.96, CFI = 0.98, and IFI = 0.97. The results showed that the model and the data fit well, and the multidimensional emotional regulation questionnaire had good structural validity.

#### 2.2.2. Emotional Regulation Self-Efficacy Scale

The scale is measured by 17 items ([11]), of which 6 items measure the self-efficacy of expressing positive emotions (e.g., “I will express happiness heartily when attending a party”), and 11 items measure the management of negative emotion self-efficacy (e.g., “I will avoid getting annoyed when people deliberately pick on me”). The scale is on a Likert scale of 6. Higher scores indicate stronger regulatory emotional self-efficacy. This questionnaire has good applicability in the Chinese population ([43]), and the internal consistency coefficient for this study was 0.81. The results of the confirmatory factor analysis showed that *χ*^2^/*df* = 2.94, RMSEA = 0.05, GFI = 0.89, AGFI = 0.86, TLI = 0.91, CFI = 0.92, and IFI = 0.92. The results showed that the model and the data fit well, and the multidimensional emotional regulation self-efficacy scale had good structural validity.

#### 2.2.3. Beck Depression Scale

The second version of the Beck Depression Scale ([8]) was used, and it obtained good reliability and validity in the Chinese college student population ([46]). The scale measured the severity of the depression of the participants over two weeks and included 21 items; each of the 21 items was scored on a Likert 4 scale. Higher scores indicate more severe depression. In this study, the internal consistency coefficient of the questionnaire was 0.67. The results of confirmatory factor analysis showed that *χ*^2^/*df* = 3.00, RMSEA = 0.05, GFI = 0.88, AGFI = 0.85, TLI = 0.91, and CFI = 0.92, IFI = 0.92. The results showed that the model and the data fit well, and the Beck Depression Scale had good structural validity.

### 2.3. Procedures

The University Ethical Committee gave its approval for the project. Before the experiment, all the participants signed an informed consent form, and each participant was given a USD 3 test fee upon the completion of the experiment. The research procedures included four stages: material preparation, participant recruitment, pretest, and intervention and post-test (see Figure 1). 

### 2.4. Preparation of Experimental Materials

In order to prepare the reading materials for the intervention group, the actual situation of the college students who used CR strategies was investigated using an open-ended questionnaire before the intervention. In order to ensure the validity of the formal experiment, the individuals who took part in the open-ended survey were prohibited from joining the formal experiment. Additionally, they were instructed to keep the details of the experiment to themselves and not share any information with others.

First, to develop the initial version of the open-ended questionnaire, several trained psychology undergraduates meticulously reviewed the existing literature and engaged in several discussions. Then, psychology undergraduates who were not involved in the preparation of the questionnaire were invited to review, provide feedback on, and suggest modifications to the questionnaire. After making the necessary amendments, we invited teachers majoring in psychology to validate the revised questionnaire. Finally, we invited college students who did not major in psychology to take the assessment to ensure that the questionnaire was clear and easily understood.

The following was the specific structure of the final open-ended questionnaire:(1)To understand the events and sources that cause college students’ negative emotions, we asked, for example, “Think back over your college life. What were the three things (academic/emotional/interpersonal) that caused you to feel or experience (sadness/pain/depression)? Please be as detailed as possible”.(2)Some questions asked the participants to write about their experiences and feelings regarding specific events and how they successfully mediated their negative emotions. For example, “When you recall this incident, did you think it was your own problem or the problem of external people or things? Please write down what you were thinking. To make yourself feel better, did you look at the situation from a different angle and convince yourself that it was not as bad as you thought it was? Did you feel better after doing this? Please provide enough details regarding how you convinced yourself or how you changed your mind so that other students can learn from your experience”.(3)To encourage college students to adopt positive emotional regulation strategies, they were asked to provide advice for themselves. For example, “When something similar happens again, will you just suppress your emotions or try to adjust your emotions? Why? If you were to give advice to a college student going through the same situation, what would you say to that student? Your words may give them ideas to improve their mood and help them overcome difficult situations and negative emotions”.

We sent the constructed questionnaire to twenty college students (ten male students and ten female students). Based on the feedback from the participants, we incorporated them into the intervention materials with CR strategies, which subsequently served as the reading resources for the intervention group throughout the intervention.

The final reading materials of the experimental group included “The main source and feelings of the last group of college students’ negative emotions” and “What did the last group of college students want to say to you?”.

Specifically, “*The main source and feeling of the last group of college students’ negative emotions*” included examples of college students successfully adjusting their negative emotions through CR strategies. For example,


*“After I entered college, I was sad to find that I had drifted apart from my former friends. We went to college in different places, were exposed to different environments and people, and gradually, we went from sharing everything to having nothing to say. At first, I was always sad and depressed, but later I told myself that true friendship does not change because of time or distance, maybe he is just not the right person. Some things cannot be changed; life will not be stagnant because of a person; rather than repressing their own heart, it is better to re-understand, and I am more in tune with friends. Now I have several good buddies, and I’m very thankful for who I was then… I was not consistently immersed in a depressed atmosphere; however, I did actively adjust my emotional state and proactively embraced a new life”.*


“*What the last group of college students wanted to say to you*” could strengthen participants’ acceptance and recognition of their own situation and further change their own construction of events. For example,


*“The most effective method for avoiding the negative effects of emotional distress is to maintain a healthy lifestyle. There are so many positive moments in your life. Do not dwell too much on the shadows of the past. Try to think about a problem from different angles, and you will be enlightened”.*


### 2.5. Participant Recruitment

For the recruitment process of the subjects, firstly, we distributed the questionnaire to college students through the online platform (https://www.wjx.cn/ (accessed on 10 June 2021)). During this process, the participants could voluntarily leave their contact information for the subsequent experimental research; in the recovered valid questionnaires, the participants who met the BDI criteria were contacted by the researchers who did not participate in the formal trial. With the voluntary and informed consent of the subjects, they were selected as participants in the experiment.

### 2.6. Pretest

The pretest gathered the participants’ demographic information (including their sex, age, and grade) and three core variables of the study (including emotional regulation strategies, depression, and regulatory emotional self-efficacy).

### 2.7. Intervention

Using a double-blind experimental design, the participants were divided into an intervention group and a control group at random. The duration of the intervention was approximately 30 min. Neither the specific hypotheses being investigated nor the differences in the experimental conditions were known to the participants in either group.

The experimental group received intervention materials, namely, the CR of emotional regulation strategy reading materials, and the control group received blank intervention materials, namely, autonomous vehicle materials. The length and reading duration of the materials were the same in both groups.

After reading the materials, the subjects were requested to engage in the related writing assignments, drawing from their personal experiences. “*Please consider your own experiences. When you encounter similar difficulties or failures and feel depressed and sad, how do you change your view of things to alleviate your sadness? Please write it down, including the whole process and psychological changes (approximately 300 words are required). With your consent, we will select a section of the composition and anonymously display it to other college students who may experience the same difficulties, thus providing them with ideas to improve their emotions and help them overcome difficulties and negative emotions. I am sure they will appreciate your efforts*”.

This practice was used to encourage the subjects to “*consider themselves benefactors rather than beneficiaries*,” to infuse a sense of purpose into their involvement in the experiment and enhance both their engagement and effectiveness. A similar approach has worked well in previous studies ([33]). The participants in the control group answered the short-answer questions after reading the materials. To avoid the excessive consumption of cognitive resources and fatigue, the short-answer questions were either underlined or prompted at the start of the obvious paragraph. This procedure successfully mitigated the impact of extraneous variables within the experiment, removed the general influence of the intervention form (reading, writing, and expression), and allowed the unique role of the intervention content (CR intervention) to be examined.

### 2.8. Post-Test

The core variables of this study (including emotional regulation strategy, depression, and regulatory emotional self-efficacy) were measured, and the same scale that was used for the pretest was used.

### 2.9. Data Analyses

Before the data analysis, the normality test of the data was first carried out, and the result showed that all the data were normally distributed, so the following data analysis was carried out. The data analyses proceeded in four steps. First, SPSS 21.0 was used to conduct a Pearson correlation analysis, and the results showed that gender was negatively correlated with expression inhibition (*r* = −0.22, *p* < 0.05) and was negatively correlated with regulatory emotional self-efficacy (*r* = −0.32, *p* < 0.01), so gender was used as a control variable in the subsequent experimental analysis. Second, SPSS 21.0 was used to calculate and analyze the pretest and post-test experimental data of the intervention group and the control group. Third, a repeated measures analysis of variance was computed via SPSS 21.0 to examine the intervention effect of the online cognitive reassessment intervention. Finally, the PROCESS Macro (Model 4) was used to test the mediating effect of regulatory emotional self-efficacy ([20]). This study was a mixed experimental design, and all the inter-subject factors were operational factors. Therefore, according to the effect size standard proposed by Cohen, 0.2 was a small effect size, 0.5 was a medium effect size, and 0.8 was a large effect size ([12]).

## 3. Results

### 3.1. The Homogeneity Analysis of the Variables in the Pretests of the Two Groups

The independent sample *t*-test was used to test the homogeneity of the main variables of the two groups in the pretest. The results are displayed in Table 1.

Table 1 shows that before the intervention, the data of two groups were homogeneous. This shows that the homogeneity of the two groups of subjects was good and that it conforms to random grouping.

### 3.2. Analysis of the Intervention Effect

To investigate the intervention effect of the online cognitive reassessment intervention, groups (intervention/control) were used as between-group variables, and time (pretest/post-test) was used as the within-group variable. A two-way mixed ANOVA was used to investigate the changes in cognitive reassessment, expression inhibition, regulatory emotional self-efficacy, and depression levels. The results are displayed in Table 2.

The results for cognitive reappraisal revealed that the main effect of time was significant, *F*(1, 96) = 8.20, *p* = 0.005, *η*^2^*_p_* = 0.08. That is, the intervention strategy in this study could improve the utilization rate of the cognitive reappraisal strategy in depressed college students.

Furthermore, the results for depression intervention revealed that the main effect of the groups was significant, *F*(1, 96) = 11.03, *p* = 0.001, *η*^2^*_p_* = 0.10. The depression score for the intervention group (*M* = 18.24, *SD* = 10.61) was significantly lower than that of the control group (*M* = 25.67, *SD* =10.51). The main effect of time was also significant, *F*(1, 96) = 11.22, *p =* 0.001, *η*^2^*_p_* = 0.11. The depression score on the pre-test in the intervention group (*M* = 24.47, *SD* = 4.34) was significantly higher than that on the post-test (*M* = 18.24, *SD* = 10.61), and the interaction effect between the groups and time was significant, *F*(1, 96) = 9.34, *p* = 0.003, *η*^2^*_p_* = 0.09.

After a simple effect analysis, the intervention group’s post-test depression score was noticeably lower than its pretest score, *F* (1, 96) = 20.51, *p* < 0.001, *η*^2^*_p_* = 0.176, and the control group’s depression scores before and after the intervention did not differ significantly, *F*(1, 96) = 0.04, *p* = 0.836, *η*^2^*_p_* < 0.001. These findings indicated that the online cognitive reassessment intervention in this study significantly reduced the depression levels of college students with depressive tendencies.

Therefore, the CR intervention materials designed in this study had significant effectiveness. Specifically, writing after reading CR intervention materials enabled the college students to adopt more CR strategies for their subsequent behaviors and significantly reduced their depression levels, thereby verifying Hypothesis 1.

### 3.3. Test of the Mediating Effect of Regulatory Emotional Self-Efficacy

According to the correlation results, we used gender as a covariate to conduct the mediation effect analysis. The results are displayed in Table 3. The total effect of the intervention conditions on depression was significant (*effect size*_Total_ = 7.43, SE = 2.13, *p* < 0.001, 95% CI = 3.21~11.65). The intervention conditions could not only directly affect the depression of college students (*effect size*_direct_ = 5.68, SE = 2.00, *p* < 0.001, 95% CI = 1.70~9.65), but also indirectly affect the depression level through regulatory emotional self-efficacy (*effect size*_indirect_ = 1.75, SE = 0.91, *p* < 0.01, 95% CI = 0.09~0.32).Thus, the intervention conditions significantly affected the level of depression, and regulatory emotional self-efficacy had a major mediating role throughout the process, so Hypothesis 2 is supported.

## 4. Discussion

This study integrated online reading and writing with college students’ daily life experiences to assist depressed students to more effectively employ CR strategies. This approach aimed to actively shape a nonthreatening framework for interpreting the challenges of daily campus life ([38]). These nonthreatening interpretations enhance individuals’ positive perceptions of campus life and modify their entrenched thought patterns, effectively reducing levels of depression through changes in both emotions and cognitive processes. This finding aligns with previous research outcomes that claim to improve the use of CR strategies that effectively alleviate depressive symptoms in college students ([37]; [42]).

One possible reason for the effectiveness of the cognitive reappraisal intervention strategies developed in this study is summarized in three points. First, according to the emotional regulation model, CR more effectively alters the course of emotional experiences than does ES. More specifically, the CR approach helps individuals perceive and express more positive emotions while reducing negative emotions ([19]), a finding echoed in numerous empirical studies that have demonstrated that CR significantly reduces negative emotions among college students ([14]; [27]). Another possible reason is that the negative stimulus-related attention transfer ability and the working memory ability of depressed individuals are impaired, making it difficult for these individuals to reshape events, thus causing them to habitually reduce their use of CR ([37]). In such cases, individuals should exercise their ability to reshape negative events to alleviate depression symptoms. Finally, a negative attention bias may also be a reason. Behavioral and neurophysiological studies have determined that an attention bias may play a role in the relationship between CR and depression ([24]). For example, as a negative attention bias may interfere with the use of adaptive strategies, intervening in the individual’s level of CR may reduce their attention bias and alleviate depressive symptoms.

This study also found that the intervention of cognitive reappraisal strategies in depressed college students can not only directly reduce the level of individual depression but also reduce the level of depression by improving regulatory emotional self-efficacy. According to the despair theory, individuals at a high risk of depression tend to perceive the causes of negative events as stable and uncontrollable ([3]). This perception makes it even more challenging for them to cope and leads to heightened negative emotions ([1]). Often, individuals adopt the emotional regulation strategy of ES, which perpetuates negative emotions rather than resolving them. These prolonged negative emotions can exaggerate the severity of challenges and foster a negative cognitive pattern in which the individual feels incapable of handling adversity, thus undermining the person’s confidence and motivation to cope effectively. Consequently, these factors increase the likelihood of depression. Intervention strategies in emotional regulation, such as CR, aim to increase the individual’s confidence to manage negative emotions and encourage the individual to confront negative experiences actively, thereby mitigating depression levels. Secondly, the CR interventions enhanced the regulatory emotional self-efficacy of depressed college students and subsequently reduced their levels of depression, which is consistent with the findings of previous studies ([2]; [7]; [47]). This result may be because that the low self-confidence of individuals with low self-efficacy in self-regulating depression and painful emotions leads them to adopt negative coping styles in the face of adverse feelings, making it difficult for them to avoid indulging in adverse emotions. This negative state then reinforces the individual’s low emotional regulation self-efficacy, and this vicious cycle eventually leads to depression ([37]; [50]).

In the simple cognitive reappraisal intervention strategy of this study, by reading the detailed experiences of peers who have experienced negative events and successfully adjusted their own emotions, subjects can adopt others’ techniques for cognitively reappraising situations to regulate emotion, thus individuals gain the ability to regulate their emotions, which boosts their confidence in their capacity to deal with difficult situations. This not only provides a new perspective in the field of depression intervention, but it saves human resources on the premise of ensuring the effectiveness of intervention and can be promoted as a new method of depression prevention and treatment. Moreover, this intervention strategy can also improve the regulatory emotional self-efficacy of depressed college students, thereby reducing their depression level, further confirming the theory of the emotional regulation process model

## 5. Limitations and Future Work

Future research should address several limitations of this study. This study was an online intervention study. Although online experiments are fast, inexpensive, and concise, online experiments also have shortcomings, such as being unable to ensure that all irrelevant variables are fully controlled, thus decreasing the reliability of the results. Therefore, subsequent studies should repeat this study in standard offline scenarios, thus providing evidence for the reliability and wide use of this research method. Furthermore, while this study revealed a significant effect of a CR intervention on depression in college students with depressive tendencies, it did not verify the sustainability of this intervention method. Hence, follow-up research should be conducted to test the long-term effectiveness of the strategy. Finally, the final number of participants in our study was less than 100 out of more than 1000 large samples. Depression was significant among the college students, but most of them were reluctant to participate in the intervention, which may be one of the reasons for the recruitment difficulties of our researchers. Future intervention studies can develop pre-intervention strategies from this perspective to improve the motivation level of depressed college students to seek help, so as to enhance the effect of depression intervention.

## 6. Conclusions

Based on the intervention materials for the CR of emotional regulation strategies, we investigated the depression levels of college students with depressive tendencies through online reading and writing and reached the following conclusions:(1)The CR intervention program developed in this study is effective. The intervention method significantly reduces the tendency toward depression among college students.(2)Regulatory emotional self-efficacy mediates the influence of CR intervention on the levels of depression of college students.

## Figures and Tables

**Figure 1 behavsci-15-00562-f001:**
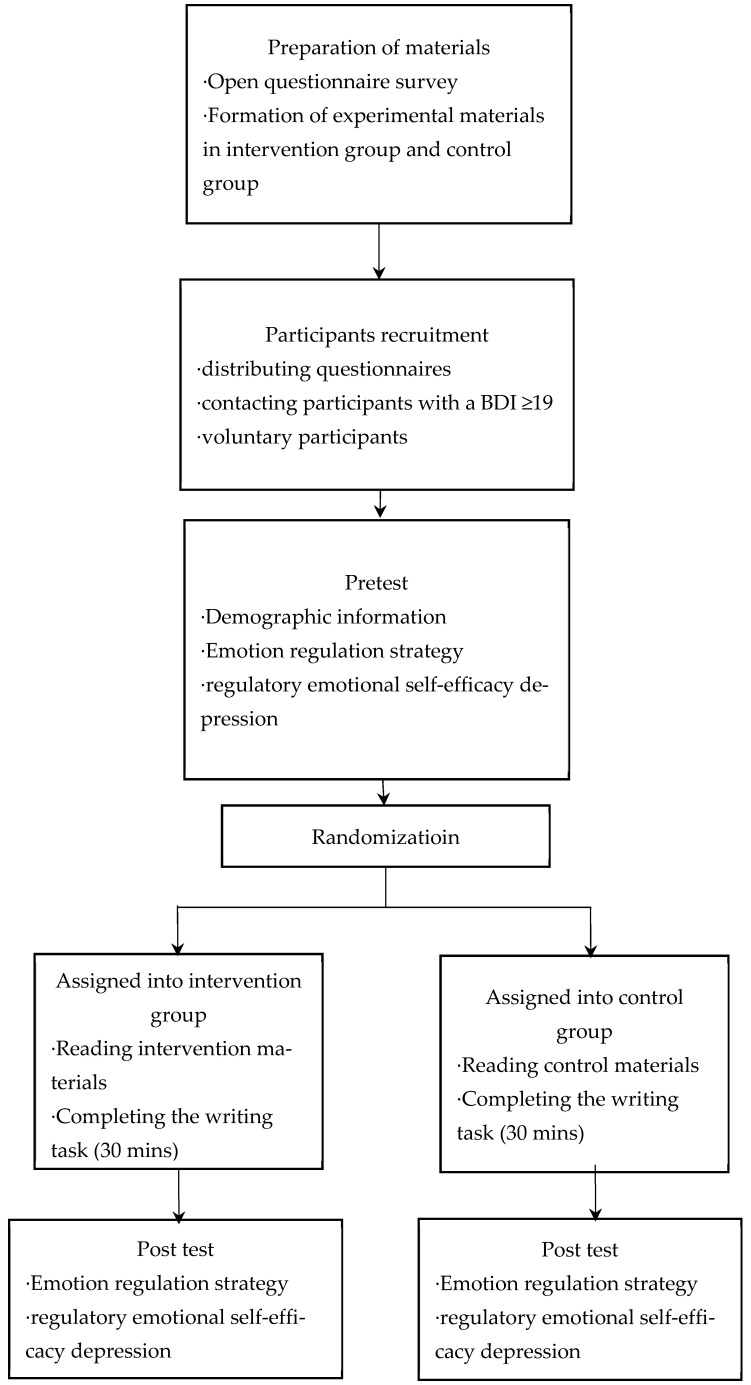
Research procedures.

**Table 1 behavsci-15-00562-t001:** The homogeneity analysis of pre-scale scores between the intervention group and the control group.

	Intervention Group (*n* = 49)	Control Group (*n* = 49)	*t*	*p*	Effect Size d
	M ± SD	M ± SD			
CR	28.94 ± 5.94	28.27 ± 6.67	0.53	0.599	0.11
ES	15.55 ± 5.74	15.61 ± 5.93	−0.05	0.959	0.01
RESE	55.04 ± 9.08	53.90 ± 8.73	0.64	0.527	0.13
Depression	24.47 ± 4.34	25.96 ± 5.24	−1.53	0.128	0.31

CR, cognitive reappraisal; ES, expressive suppression; RESE, regulatory emotional self-efficacy.

**Table 2 behavsci-15-00562-t002:** Analysis of the intervention effect.

Variables	Groups	Pretest (*n* = 49) M ± SD	Post-Test (*n* = 49) M ± SD	Time Main Effect		Group Main Effect		Time–Group Interaction Effect	
				*F*	*η* ^2^ * _p_ *	*F*	*η* ^2^ * _p_ *	*F*	*η* ^2^ * _p_ *
CR	Intervention	28.94 ± 5.94	30.92 ± 6.05	8.20 **	0.08	1.42	0.02	2.73	0.03
	Control	28.27 ± 6.67	28.80 ± 6.09						
ES	Intervention	15.55 ± 5.74	16.04 ± 5.88	1.04	0.03	0.38	0.01	0.16	0.01
	Control	15.61 ± 5.93	16.02 ± 5.09						
RESE	Intervention	55.04 ± 9.08	58.98 ± 9.73	15.06 ***	0.14	2.00	0.02	3.80 *	0.04
	Control	53.90 ± 8.73	55.20 ± 9.35						
Depression	Intervention	24.47 ± 4.34	18.24 ± 10.61	11.22 ***	0.11	11.03 ***	0.10	9.34 **	0.09
	Control	25.96 ± 5.24	25.67 ± 10.51						

* means *p* < 0.05, ** means *p* < 0.01, *** means *p* < 0.001. CR, cognitive reappraisal; ES, expressive suppression; RESE, regulatory emotional self-efficacy.

**Table 3 behavsci-15-00562-t003:** The regression analysis of the mediating effect of regulatory emotional self-efficacy.

Dependent Variable	Independent Variable	*R* ^2^	*SE*	*B*	*t*	*p*
Depression	Intervention condition	0.11	2.13	7.43	3.50	<0.001
	Gender	0.12	2.12	2.83	1.33	0.187
RESE	Intervention condition	0.14	1.93	−3.78	−2.06	0.042
	Gender		1.83	−6.18	−3.37	0.124
Depression	Intervention condition	0.27	2.00	5.68	2.84	0.006
	RESE		0.11	−0.46	−4.22	<0.001
	Gender		2.08	−0.03	−0.02	0.988

RESE, regulatory emotional self-efficacy.

## Data Availability

The datasets generated during and/or analyzed during the current study are available from the first author upon reasonable request.

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
