# Peer review of "The Impact of Cognitive Reappraisal Intervention on Depressive Tendencies in Chinese College Students: The Mediating Role of Regulatory Emotional Self-Efficacy"

_behavsci, 2025, doi:10.3390/bs15050562_

Round 1
Reviewer 1 Report
Comments and Suggestions for Authors
Please, see the attached file.

Author Response
Dear reviewers,
Thank you very much for taking the time to review our manuscript. We have answered your questions and suggestions, and made major revisions and adjustments to the manuscript. Your suggestions have greatly contributed to the professionalism of our manuscript, and we have submitted a document to answer your questions and suggestions in detail, please see the attachment. Thank you again and have a nice life.

Reviewer 2 Report
Comments and Suggestions for Authors
Thank you for the opportunity to review “The Impact of Cognitive Reappraisal Intervention on Depressive Tendencies in Chinese College Students: The Mediating Role of Regulatory Emotional Self-efficacy”, which was an experimental study in which cognitive reappraisal was found to decrease depression both directly and indirectly as a function of regulatory emotional self-efficacy. The manuscript is well-written with very intriguing findings. While I have no substantive concerns about the paper, I do offer several suggestions for manuscript improvement. I organize my comments by section below:
Introduction:
I have two comments about the intro. First, the authors begin the paper by defining and describing the variables of interest and how they are related to each other. The authors do a good job of this, though I recommend they provide a bit more transition content connecting the variables to college-aged populations. The majority of the introduction does not specify specific developmental levels for these relations, assuming that it generalizes to everyone. However, I think adding a bit of content on why these relations are uniquely predictive for college students is warranted and would strengthen the introduction.
Second, the final paragraph of the introduction attempts to pitch regulatory emotional self-efficacy as a mediator, but I do not think the authors did enough to explain why that variable serves as a mediator. The way it is written, it sounds more like a covariate than a mediator. Is there more information that could be provided as to why regulatory emotional self-efficacy might be predicted by the intervention and then why it would subsequently predict depression. The authors note that regulatory emotional self-efficacy is related to depression, so half the work is done, but I would like to see more about the a-path in this mediation model.
Methods:
Is there a possibility of selection bias in the study? The experimental sample of 98 participants was a subsample of the nearly 1000 respondents who completed the questionnaire. Is it not possible that there is something unique about college students who voluntarily chose to respond to a questionnaire on this topic? It is not a fatal flaw in my view, but something worth mentioning in the limitations section.
The Cronbach’s Alpha estimate for ERQ is a little low and should be noted as such either in that paragraph in the measures section or somewhere in the limitations.
In describing the emotion regulation self-efficacy scale, the authors note that one of the subfactors is managing negative emotions and provide the following example item: “I will feel very proud when athletes win glory for the country”. Why is this negative? This may be an issue of cultural specificity of items, with this item being perceived as negative in China. Transparently, I am from the USA, and therefore it may be ignorance on my part, but I do not see the above item as negative. Can more clarity be provided as to why this (and other) items were conceived as negative or positive? Perhaps this was from the original interviews conducted with students and I just missed. Please clarify for my own information.
The authors provide reliability coefficients in the form of Cronbach’s Alphas, which is good, but best practice in recent years is to provide further psychometric info for multi-item measures. In particular, model fit information from confirmatory factor analysis (CFA) is the gold standard of psychometric evidence for the measures included. I recommend the authors consider adding this information.
How much remuneration was provided to participants? The procedures section does not specify this.
Components of the intervention remind me of other brief interventions in which students see testimonials from past students (Rosenzweig et al., 2020; Stephens et al., 2014). The authors could consider citing these papers to reinforce the benefits of their approach:
Rosenzweig, E. Q., Wigfield, A., & Hulleman, C. S. (2020). More useful or not so bad? Examining the effects of utility value and cost reduction interventions in college physics. Journal of Educational Psychology, 112(1), 166–182. https://doi.org/10.1037/edu0000370
Stephens, N. M., Hamedani, M. G., & Destin, M. (2014). Closing the social-class achievement gap: A difference-education intervention improves first-generation students’ academic performance and all students’ college transition. Psychological science, 25(4), 943-953.
The authors note that none of the demographic variables were correlated to the four main variables and therefore none of the variables were included in the experimental analysis. My first question is whether or not the demographic variables were related to the four variables both pre and post intervention? There should be multiple points of data for at least a few of those variables, so are none of the demographic variables correlated to the variables at either point? More importantly, bivariate correlations are not, to me, a strong enough indicator of relations to justify variable removal. Since more sophisticated analyses control for other variables included in the model (certainly the mediation analysis), I am not sure that looking at correlations is enough to justify taking out variables. I would either a) Provide more justification for not including demographic as covariates or b) Include demographics as covariates.
Results:
Why did the authors conduct a series of paired samples t-tests rather than doing a repeated measures MANOVA? The multiple DVs resulting in multiple data runs raises the risk of type-1 error inflation, especially for a small sample size of n = 98. I recommend the authors consider this alternative approach.
The analysis of the intervention effect section states that a repeated measures ANOVA was conducted, but it is actually a mixed ANOVA since there is a between-groups and within-groups component.
Discussion:
The first paragraph could be moved to the results section since it mentions hypothesis 1 was supported. Text on hypothesis 2 being supported was included at the end of the results, and I recommend content on both hypotheses being supported be included in the results section. The way the discussion is worded, the whole first paragraph could be lifted and moved to the results and then the discussion could start with the second paragraph “This study integrated online reading…”. This is just my suggestion, but I do think that both hypotheses should be mentioned in the results rather than one in results and one in discussion.
The authors note in the limitations that the online environment was suitable for the pandemic. Was the pandemic relevant to the implementation of the study? Alternatively, could there be a cohort effect of the pandemic that might affect these results? I recommend the authors either a) Not mention the pandemic in their discussion or b) Explain how or why the pandemic may have affected the results, or not. Either way, the authors should include when the data collection occurred in the methods section, so a reader knows how soon after the pandemic this study occurred. This would help determine how disruptive the pandemic may have been to this study and the authors’ conclusions.
Grammatical Edits:
The following are grammatical edits throughout the paper that should be corrected. I encourage the authors to review the entire paper for grammar just to be sure there are no more errors.
Page 1, line 11: The abstract states “regulation emotional self-efficacy”. Is it not “regulatory” emotional self-efficacy.
Page 2, line 55: Unclear sentence: Emotion regulation describes how people control their emotions arise and manifest themselves
Page 3, line 109: Typo detected: “Likert7”
Page 3, line 125: Typo detected: “Likert4”
Page 4, line 173: Oxford comma should be added after “feedback”
Page 7, line 299: “which then significantly” implies causal order between cognitive reassessment strategies and depression, and paired samples t-tests cannot do that. I recommend revising this.
Page 8, line 321: “Tables 4” should be singular and not plural
Page 8, line 327: “Moderating” should be “mediating”
Page 9, line 337: “Employing” should be “employ”
Page 9, line 343: “effectively alleviates”should be “that effectively alleviate”
Page 9, line 345: First sentence is incorrect. “One possible reasons for this result are summarized in three points.” should be “one possible reason for this result is summarized in three points”
Overall, the paper will be a great contribution to the literature, and I applaud the authors on their efforts.
Author Response

(The authors gave the same response as above.)

Round 2
Reviewer 1 Report
Comments and Suggestions for Authors
accept in this form
Author Response
Dear reviewers,
Thank you very much for taking the time to review our manuscript again. Wish you have a nice life.
Reviewer 2 Report
Comments and Suggestions for Authors
Thank you for the opportunity to re-review “The Impact of Cognitive Reappraisal Intervention on Depressive Tendencies in Chinese College Students: The Mediating Role of Regulatory Emotional Self-efficacy”, which was an experimental study in which cognitive reappraisal was found to decrease depression both directly and indirectly as a function of regulatory emotional self-efficacy. The authors have done an admirable job addressing all of my feedback. The only remaining item that I see is that the authors still refer to their analysis as a two-way repeated measures ANOVA when I am pretty sure it should be called a two-way Mixed ANOVA since there is both a between and within component. Other than that, I have no further reservations about this manuscript. I congratulate the authors on their contribution to the literature.
Author Response
Dear reviewers,
Thank you very much for taking the time to review our manuscript again. Based on your suggestions, we have made the following modifications:
Comments: "The only remaining item that I see is that the authors still refer to their analysis as a two-way repeated measures ANOVA when I am pretty sure it should be called a two-way Mixed ANOVA since there is both a between and within component."
Response: We have changed "a two-way repeated measures ANOVA" to "a two-way Mixed ANOVA" in the manuscript to conform to more scientific and standardized academic terms.
Thank you again and have a nice life.